# Dementia Education for Workforce Excellence: Evaluation of a Novel Bichronous Approach

**DOI:** 10.3390/healthcare12050590

**Published:** 2024-03-05

**Authors:** Leah Macaden, Kevin Muirhead

**Affiliations:** 1Nursing Studies, School of Health in Social Science, University of Edinburgh, Edinburgh EH8 9AG, UK; 2Department of Nursing & Midwifery, University of the Highlands and Islands, Inverness IV2 3JH, UK

**Keywords:** dementia education, blended learning, online learning, bichronous learning, mixed methods, evaluation, workforce

## Abstract

Dementia education and training for workforce development is becoming increasingly important in bridging knowledge gaps among health and social care practitioners in the UK and internationally. Dementia Education for Workforce Excellence (DEWE) was developed during the COVID-19 pandemic, blending both synchronous and asynchronous instruction and delivered across three different contexts: care homes, home care, and nurse education within the UK and India. This study aimed to evaluate DEWE using mixed methods with online survey data analyzed descriptively and interview data analyzed thematically. Integration of survey and interview data aimed toward a comprehensive evaluation of this novel approach for dementia workforce development. Thirty-four social care practitioners and nurse educators completed the online survey demonstrating high-level learner satisfaction, learning gains, behavioral change, and motivation to share new knowledge. Four key themes developed from the analysis of interviews (n = 9) around participants’ pursuit of new knowledge; delivery modes in DEWE; learning gains and impact of DEWE; and adaptations for future program implementation. Findings suggest DEWE is an innovative resource that promotes person- and relationship-centered dementia care across all stages of one’s dementia journey. Cultural adaptations are recommended for international delivery to ensure contextual alignment and maximum impact.

## 1. Introduction

There are concerns internationally around the knowledge, skills, and attitudes of health and social care practitioners who provide care for people living with dementia [1]. Globally, the prevalence of dementia is 55 million with projections indicating an increase to 139 million by 2050 [2,3]. Notably, most of these people, approximately 75%, are anticipated to be concentrated in low- and middle-income countries [4].

In India, dementia prevalence for people aged 60 years and older is estimated to be 7.4% and increasing due to demographic transition [5]. Despite this, there is limited awareness and stigma associated with the condition [1]. Dementia symptoms can be considered as non-pathological deviations from normal ageing and supported within systems of family care with limited input from professionals [6]. The Indian Government has recognized the challenges with special provision for people with dementia included in national health programs and policy [1]. The Dementia India Report [7] highlighted health professional education as a critical component when developing national dementia services. Key areas for change included the development of in-service training for health professionals and Train the Trainer programs for carers and community health workers.

In the UK, dementia workforce development through education has been articulated in several government reports and strategies [8,9,10,11,12,13,14]. Dementia education in adult social care is a priority to meet the needs of practitioners who are increasingly providing specialist dementia care. Evidence suggests that asynchronous online courses can benefit social care practitioners in terms of knowledge and skills development [15,16]; however, social distancing requirements during the COVID-19 pandemic highlighted the value of integrating both asynchronous and synchronous delivery methods [17]. No studies are known to have combined asynchronous and synchronous online dementia education for social care practitioners, nor are there any studies known to have evaluated technology-enabled dementia education in India.

### 1.1. The Intervention 

Dementia Education for Workforce Excellence (DEWE) was developed during the COVID-19 pandemic to equip health and social care practitioners to work in increasingly digitalized and ageing societies with growing numbers of people with dementia. DEWE integrates the philosophy of care within the Standards of Dementia Care in Scotland [11] that are underpinned by three National Dementia Strategies [10,12,13]. Program development was informed by Scotland’s national dementia workforce development framework (Promoting Excellence: A framework for all health and social services staff working with people with dementia, their families, and carers) [14]. This framework prioritizes the rights of individuals living with dementia and integrates their dementia journey in its framework, structured at four practice levels: Informed, Skilled, Enhanced, and Expert. The levels correspond to the frequency and intensity of interactions that practitioners engage with people affected by dementia within their unique practice settings. DEWE was aligned to the skilled level of Promoting Excellence which outlines the knowledge and skills required by practitioners who have direct and/or substantial contact with people living with dementia.

The project lead previously developed two innovative and comprehensive dementia curricula for pre-registration nurse education in Scotland: Being Dementia Smart [18] and Dementia Enhanced Education to Promote Excellence (DEEPE), both mapped at the enhanced level of the Promoting Excellence Framework. DEWE was developed adopting a similar approach with input and consensus from an expert group that consisted of academics, clinicians, educational technologists, and experts with lived experience of dementia or dementia care giving. 

DEWE lends itself to bichronous online learning—a blend of both asynchronous and synchronous delivery methods [17] where participants can participate flexibly during the asynchronous parts of the course alongside real-time participation for the synchronous sessions. For the asynchronous component, participants had free access to three online workbooks (Dementia Care Essentials; Dementia Care Priorities; & Dementia Care Enablers) with content mapped along the stages of the dementia journey. The key narrative was based on ‘Barbara’s Story’ [19], a filmed ethnodrama that demonstrates the various stages and associated complexities of the dementia journey supported by learning activities and opportunities for reflective discussions. Learning resources also included short videos created by a general practitioner as an expert with lived experience of dementia [20] providing an emic perspective with insights into the higher needs of people with dementia including social activity, sense of belonging, self-esteem, and meaning in life [21]. Furthermore, a relationship-centered approach to dementia care was embedded to emphasize the importance of support for the wellbeing of both formal and informal carers to ensure high-quality person-centered dementia care [22]. Synchronous sessions (n = 16), via the Cisco Webex Platform, were spread over 6 weeks and facilitated by experts (with lived experience of dementia care giving and professionals). Live sessions embraced the philosophy of person-centered dementia care regularly interspersed with opportunities for reflective learning on dementia care from pre-diagnosis to end-of-life (Table 1). 

### 1.2. Participants in DEWE 

DEWE training was provided to nurse educators and social care practitioners (including care home and home care practitioners) to cascade the training for nursing capacity building in dementia care. The training was conducted in two cohorts during the pandemic—one from the UK and another from India. The inclusion of participants from India aimed to address the considerable resource gap for dementia workforce development within the Indian context despite the increasing prevalence of dementia. 

### 1.3. Aim and Objectives 

The overarching aim of the study was to evaluate DEWE training delivered using a bichronous approach with practitioners and educators from three different contexts.

To explore participants’ perceptions on the content and quality of the dementia training.To explore participants’ perceptions and experiences of accessing the dementia training using a bichronous approach.

## 2. Materials and Methods

The evaluation was underpinned by Kirkpatrick’s Model [25] using a mixed methods approach combining quantitative and free-text survey response data with qualitative data from interviews.

### 2.1. Sampling and Recruitment 

All DEWE participants (n = 55) were invited to join the study via email, which included participant information and the contact details of the research team. Additionally, they were provided with a secure link to access and complete an online survey. Those who completed the survey had the opportunity to express an interest in participating in a one-to-one interview through a separate survey link. Interested participants were subsequently contacted by a member of the research team to schedule interviews.

### 2.2. Data Collection 

Quantitative and free-text response data were collected via the online survey platform [26] six months after participation in DEWE using a variety of Likert scale, rating scale, and open-ended questions. The survey was structured according to Kirkpatrick’s model which consists of four levels: (1) reactions to the training; (2) learning gains (knowledge, skills, and attitudes) due to the training; (3) behavioral changes due to the training; and (4) the results achieved due to the training. Consent was gained from all participants using the online survey platform. 

Semi-structured interviews were conducted virtually via the Cisco Webex platform between January and February 2023 (within 18 months of participation in DEWE). The interview guide is available as Appendix A. Consent for interviews was obtained electronically from all participants using email. The average interview duration was 33 min.

The evaluation was structured at two time points primarily to assess the effectiveness of the intervention both in terms of knowledge and the application of learning in practice. Questionnaire completion was at six months to avoid any immediate recall bias and interviews were scheduled at 18 months to allow time and space for participants to report on how their learning had informed their practice or not. 

### 2.3. Data Analysis 

Quantitative survey data were analyzed using descriptive statistics in Microsoft Excel 2019 version 2108 and reported as either frequencies, percentages, or median values. Free-text responses were analyzed in Microsoft Excel using a deductive approach for alignment with Kirkpatrick’s model and displayed alongside related quantitative data as participant quotations or in frequency tables. 

Interview data were subject to reflexive thematic analysis [27] using an inductive approach. Analysis comprised six phases: (1) interview recordings were transcribed verbatim by a member of the research team (KM) and then transcripts were reviewed multiple times for data familiarization; (2) data were coded by one member of the research team (KM) using NVivo; (3) initial codes were clustered into provisional candidate themes through a process of development, revision, and refinement; (4) candidate themes were reviewed in relation to codes and the entire data set to consider alternative options for pattern development; (5) a detailed thematic framework was produced with descriptions of themes and subthemes; and (6) an analytic narrative was created including data extracts to tell the story of the data. Analytical processes were discussed regularly by KM & LM and an audit trail maintained to enhance trustworthiness [28]. 

Findings from both the survey and interviews were reviewed, and the most salient points were extracted. These findings were then converged with equal priority given to survey and interview findings. The integration resulted in two overarching themes: strengths of DEWE and areas for development. The mixed methods results were tabulated and categorized according to Kirkpatrick’s model to facilitate the interpretation of the main study findings.

## 3. Results: Survey Findings 

### 3.1. Demographics 

Participants were 34 nurse educators and social care practitioners from the UK and India who completed DEWE between July 2021 and March 2022. Educators were either professors (n = 3) or associate/assistant professors in nursing (n = 8), nursing lecturers (n = 3), or nursing tutors (n = 3). Practitioners were carers (n = 7) or service managers (n = 2). Eight participants reported their role as being ‘other’ but did not provide any further details on their role. 

### 3.2. Experience of Dementia Care and Education 

Twelve educators (70.6%) taught on BSc Nursing programs, two (11.8%) taught on general nursing programs, whilst one (5.9%) taught on postgraduate programs. The remaining educators (11.8%) taught on a combination of programs. Five practitioners (55.6%) provided care to more than 15 people with dementia on a typical workday, whilst two (22.2%) provided care to between 11 and 15 people. Two practitioners (22.2%) did not provide dementia care at work. 

### 3.3. Access to DEWE 

A combination of personal computers and smartphones were most frequently used to access DEWE. Figure 1 demonstrates the devices used to access the training. Note the total does not tally to 34 (most likely as some participants have used multiple devices to access DEWE).

### 3.4. Level 1: Reaction to Asynchronous Resources 

The presentation of the workbooks was highly rated among participants with most indicating the color scheme, font style, font size, icons, and illustrations being very good or excellent (Figure 2). There were high levels of agreement that the online workbooks were intuitive; however, some participants suggested that the workbooks were not easy to navigate (Figure 3). One participant suggested that the content could be more streamlined as ‘*a few videos were mixed up*’ [80908493, Practitioner]. 

The workbooks were thought to work well but ‘*did not translate to hard copy which some staff without IT skills preferred to use*’ [92124189, Other]. Another participant suggested that ‘*ease of use PDF copies would be useful*’ [81022848, Practitioner]. 

Twenty-four participants rated the overall quality of the workbooks on a scale of 1–10 (1 being poor and 10 being excellent). The median scores (with interquartile range) for the three workbooks were: 9 (9, 10), 10 (9, 10), and 9 (9, 10) for Workbooks 1, 2, and 3 respectfully. The participants’ reaction to the training was also measured as levels of agreement to various aspects of the training content (Figure 4).

### 3.5. Level 1: Reaction to Synchronous Resources 

Feedback on the duration of the interactive sessions was mixed with some participants recommending a shorter duration whilst others recommended a longer duration. Alternative suggestions were to reduce the duration of interactive sessions and increase the frequency of sessions provided. Some participants were not satisfied with the delivery platform and recommended that this be changed for future delivery. Other suggestions to improve synchronous sessions included implementation of interactive icebreaker sessions, improved orientation to workbooks, and follow-up sessions. 

### 3.6. Level 1: Reaction to Bichronous Learning 

Participants described the aspects most and least enjoyed about blended learning in DEWE (Table 2). 

### 3.7. Preferred Mode of Learning 

Twenty-four participants rated their preferred modes of dementia training on a scale of 1–5 (5 being most preferred and 1 being least preferred). Figure 5 shows the median scores for each training modality with high levels of preference for a blended approach.

### 3.8. Level 2: Learning Gains

Participants rated the relevance of the learning outcomes to their professional practice on a scale of 1 to 5. The median score for each outcome was 5, indicating high levels of relevance. Additionally, there was strong consensus that DEWE had enhanced various aspects of dementia care knowledge (Figure 6). Moreover, an overwhelming majority of participants (91.2%) reported that DEWE had equipped them with relevant skills transferable beyond the dementia care context (Figure 7).

### 3.9. Level 3: Practice-Based Behavioral Change 

There were high levels of agreement that learning in DEWE had resulted in positive behaviors in practice (Figure 8).

### 3.10. Level 4: Results 

Following DEWE, participants were motivated to lead or influence dementia practice. Several were interested in leading dementia education through teaching and the development of educational initiatives: 


*“I would like to influence the way nurses learn and practice dementia care”.*
[93011072, Educator]

Participants were also motivated to share knowledge gained and improve awareness and attitudes towards the condition: 


*“I would like to help change the attitudes of some carers and give them a confidence boost and help them with their knowledge”.*
[81126963, Practitioner]

The training motivated participants to lead improvements in person-centered dementia care, communication with people living with dementia, and support for family members and carers affected by dementia. Specific areas of dementia practice that participants sought to change included dementia assessment, delirium awareness, management of stress and distress, implementation of dementia inclusive environments, pain management, and palliative care. 

## 4. Results: Interview Findings 

### 4.1. Demographics 

The participants were six nurse educators (lecturers/tutors and coordinators of nursing programs) and three practitioners with experience of health and/or social care from India (n = 7) and the UK (n = 2). Participants were issued a code reflecting their demographic profile. The first part of the code was an arbitrary number of participation; the second reflected the participants’ areas of practice (Educator = 01; Practitioner = 02); and the third related to participants’ location (India = 01; UK = 02). 

### 4.2. Themes 

Four themes developed from the data analysis, which related to: the participants’ pursuit of new knowledge; educational delivery modes in DEWE; learning gains due to DEWE; and adaptations recommended for future programs. Each theme comprised subthemes, as shown in Figure 9.

### 4.3. Theme 1: In Pursuit of New Knowledge 

#### 4.3.1. Dementia Knowledge and Learning Experiences 

Participants from India often reported superficial knowledge about dementia [P3-02-01; P4-01-01; P5-01-01; P8-02-01] and limited prior experience of dementia education [P2-01-01; P3-02-01; P6-01-01]. Despite this, there was awareness of the increasing dementia prevalence in India and limitations in national dementia surveillance systems and health programs. 


*“Dementia is also a very important disorder that is on the rise, you know, like, more and more elderly patients are having this disorder and because of that I think, yeah, we should be really into understanding more about dementia, learning more about dementia and all that”.*
[P4-01-01]

Participants were critical of dementia education in India which was limited in undergraduate nursing curricula with basic content considered inadequate to support future practice. 


*“It is just a very small, small, minute part of the curriculum which, by the time the student becomes the staff, they would have forgotten, they wouldn’t even sometimes remember I feel”.*
[P4-01-01]

Dissatisfaction with professional and vocational development in dementia care resulted from training using in-person and didactic delivery methods. 


*“We are used to like monotonous resource person going on and on and sometimes very few programs have some participation, but there is no variety in the program, you know, live program”.*
[P5-01-01]

Among participants from the UK, dementia was described to be an unattractive career option which intensified the need for comprehensive undergraduate dementia education despite competing demands within existing curricula.


*“I don’t know how familiar you are with nursing curricula but they’re always very challenging to get everything into a relatively short period of time, three years is never long enough”.*
[P7-01-02]

In-house and online dementia education were common modes for professional development in the UK. Online training was often mandated and risked disengagement due to asynchronous delivery.


*“The system that we use for our staff …you log in by yourself, you watch twenty plus videos and after each video clip you have a multiple-choice question…its quite basic”.*
[P9-02-02]

#### 4.3.2. Motivation for Learning 

DEWE provided participants with opportunities to develop new dementia knowledge which participants valued for their own professional development but also to pass on to others.


*“I’ve been a caregiver for last 20 years now to my in-laws and three years ago my mother-in-law suffered a stroke and then she had an onset of dementia…and also at [workplace] we focus a lot on senior citizens care, that is one of our main areas of work, so, I thought both professionally and personally this is going to give me a lot of insights”.*
[P8-02-01]


*“As an educator, always keen to try to enrich any students learning and, so, ……it was really good to sort of develop my own learning and understand things that are really helpful in others, in helping others to learn as well, you know, around dementia”.*
[P7-01-02]

The expert facilitation in DEWE provided a sense of credibility, which motivated learner engagement. Motivation for learning was also enhanced as the course was delivered free of charge with certification of completion particularly valued by participants from India [P5-01-01; P6-01-01]. 

### 4.4. Theme 2: A Time and a Place for DEWE

#### 4.4.1. Time-Independent Learning 

The online workbooks allowed for time-independent learning with opportunities for participants to assimilate and reflect on new concepts. The resources provided participants with ongoing access to information which could be revisited and stored with the aim of supporting learning.


*“I just found them really helpful and the fact that you can then revisit things because, you know, because my memories good at some points but others it isn’t… being able to revisit it and you could download some of the pages from that…which again was just helpful”.*
[P7-01-02]

The use of active learning strategies including quizzes and games supported information retention. Real-life and emotive videos provided a proxy for experiential learning and resulted in emotional engagement which supported information retention and development of compassionate care and empathy for people with dementia.


*“Seeing videos like that, you’re seeing it from the other person’s point of view…I understand now that they can’t help it, you know, it’s confusing for them”.*
[P9-02-02]

A potential barrier to self-directed learning was self-motivation for engagement. Additional barriers included isolation during learning, which was perceived problematic in the context of dementia education.


*“I don’t think you learn in the same way when something is strictly online…a topic like dementia is about people, and you almost want to test out, sort of, you know, your own learning with others”.*
[P7-01-02]

#### 4.4.2. Learning in Real Time 

Synchronous sessions made learning more interesting through group interactions with peers, facilitators, and guest speakers, allowing participants to gain knowledge from diverse experiences and perspectives. Learning was enriched from the cultural and professional diversity within the learning groups and opportunities to learn from a variety of professional and personal experiences. 


*“Like it says like ‘none of us is smarter than all of us’, so, it’s like the years of wisdom comes in a room and all that, all those years of experience and learning, so that was very rich”.*
[P1-01-01]

Sessions involving people with lived experience of dementia resulted in depth of emotional engagement and an appreciation of the realities of informal dementia care. 


*“Lived experiences of people was more encouraging and practical and to learn from their life experiences, so that was very good and that was really touching our hearts”.*
[P5-01-01]

Participants were motivated and inspired by the facilitator/s who contributed to live sessions. 


*“I just thought she was amazing, so much passion, and you could tell that she loved making the lives better for the people that they had living with dementia, and it was just, yeah, it was heart-warming”.*
[P09-02-02]

The weekly sessions provided consistent facilitation and opportunities to build on prior learning. The synchronous platform enabled active group learning whilst simultaneously encouraging introspection and personal reflection.


*“We had lots of opportunities to understand our perspectives as well as other perspectives, like it was a platform for us to introspect whether our understanding or our perspectives about dementia is right or what else I can look for when it comes to the care of patients”.*
[P6-01-01]

Barriers to synchronous sessions included poor online etiquette and time limitations, which would require skillful moderation and opportunities for all participants to be heard.


*“I think that is just a fact of, you know, learning this way…obviously we encounter it all the time…there’s usually somebody who’s quite vocal, somebody else is trying to talk, but because you can’t see them you aren’t really aware of that, so that to me was probably a drawback”.*
[P7-01-02]

Participants from India were less accustomed to the WebEx platform and occasionally experienced connectivity issues and difficulty joining synchronous sessions. 


*“We did have challenges using Webex because many times our people were not able to join, we are so accustomed to Zoom, that was the first time I think many of us were using Webex”.*
[P5-01-01]

#### 4.4.3. Bichronous Learning 

Participants considered the bichronous delivery to be a novel approach using multiple methods to meet the needs of a diverse audience and learning styles. 


*“It was actually perfect blending or perfect package of all three components, like all three domains of teaching, affective, cognitive, and psychomotor…so, it’s a perfect blending of all three domains of teaching”.*
[P6-01-01]

The asynchronous resources enriched participation in synchronous sessions from prior preparation. Correspondingly, synchronous sessions elaborated on asynchronous content. This reciprocity seemed to influence enhanced learning.


*“I was able to take away much from the class because I was able to read and then come to the class, otherwise I don’t think it would have been so much of beneficial to me”.*
[P8-02-01]

Benefits of the bichronous approach also included the potential to revisit/reuse asynchronous resources and recordings of synchronous sessions as often and for as long as required to support learning. 


*“We can go over and over whenever we have a doubt, we can go back and learn from them, so the freedom to access that has been given to us…that is a positive thing”.*
[P5-01-01]

DEWE had become more relevant since the COVID-19 pandemic and was considered to be an enjoyable and motivating learning experience. 


*“Because this sort of fell during the pandemic, you know, we were adapting to online, you know, very quickly anyway…we were getting so familiar with the way that we were teaching anyway”.*
[P7-01-02]

### 4.5. Theme 3: Impact from Learning in DEWE

#### 4.5.1. Learning Gains 

Aspects of new knowledge gained by participants in DEWE are shown in Table 3. 

DEWE had broadened participants’ understanding about dementia and also resulted in the development of positive dementia care attitudes. 


*“I have learned on how to be…an empathetic caregiver… an informed caregiver”.*
[P8-02-01]

#### 4.5.2. Impact 

Participants reported having a greater awareness of the signs and symptoms of dementia following DEWE including the behaviors of older adults in their care and communities. 


*“Earlier, it was not in my thinking, or it was not in part of my internal function, to think of something like this… to wonder if a person could have dementia”.*
[P1-01-01]

One participant who worked in clinical areas shared an anecdote of using the knowledge gained with greater confidence to support people living with dementia in practice: 


*“On the last shift I did, I actually was in a bay, you know, with six ladies and one of whom, you know, did have dementia, so, you know, constantly I was just going back, you know, reiterating what I’d said, holding her hand and everything else and it was only when I was going back to another lady and they went ‘oh we’re going to have a quiet day today’ and I said ‘oh, have things been, you know, disturbing you?’ and she said, ‘yes’ she said ‘that’s the most attention she’s had since we’ve been here’”.*
[P7-01-02]

The course provided participants with the confidence to provide informal health education and support for friends and relatives. Learning from DEWE created a positive ripple effect as participants confidently shared knowledge with others. 


*“There was something I learned, it was so profound, I put it on my social page…it was nice that some of the people within my friends circle who saw that, they replied back, and they told that, thank you so much it was a very good information and because they have somebody in their family who are living with dementia”.*
[P4-01-01]

Participants identified potential to incorporate concepts from DEWE into nursing curricula and educational resources for practitioners and dissemination on a wider scale both in India and the UK. 


*“It was a particularly big blessing for me because I’ve written like some modules on elderly care, so I could apply this into that, and see, these modules and curriculum that we’ve designed is being used to train primary health level workers, nurses across India”.*
[P1-01-01]


*“I think it should be rolled out everywhere, I think it should be like teamed up with the NHS, I think it should be done with all new staff starting, even old staff…. I think it would be a fantastic program to roll out”.*
[P9-02-02]

Several participants were motivated to engage with further learning about dementia in other online courses [P3-02-01], masters [P9-02-02], and PhD programs [P2-01-01].

The impact from DEWE also motivated participants from India to develop programs of research including timely inquiries into dementia prevalence in local communities. 


*“I started to read articles on dementia related to India and South Asian countries, I found that there were very few nursing related articles, so, a colleague and I decided to conduct a small study and we’re in the process of doing that”.*
[P3-02-01]

### 4.6. Theme 4: Adapting and Appropriating DEWE for Future Success

#### 4.6.1. Improving DEWE 

Several participants suggested that future modifications to DEWE were not necessary [P1-01-01; P3-02-01; P5-01-01; P7-01-02]. Despite benefits of the bichronous online delivery, some participants noted a preference for in-person [P1-01-01] and practice-based learning [P4-01-01].

Discussion in DEWE had focused on participants’ experience, perceptions, and perspectives of dementia care. Recommendations to improve synchronous sessions were to include case presentations based on actual experiences using a systematic approach incorporating patient assessment, evaluation, and care planning [P6-01-01]. Assessment and evaluation of learning were recommended using multiple choice questions and concept mapping techniques [P2-01-01]. Other suggestions to improve DEWE included even greater focus on neurophysiology [P2-01-01] and the dementia care environment [P6-01-01]. 

Participants recommended that DEWE is updated regularly to reflect evidence-based changes in dementia care.


*“We want this to be given again and again with some other changes or a kind of recap, recapitulation, or kind of refresher course, we want this to be at least once a year or twice a year … I feel that it has to be refreshed”.*
[P6-01-01]

#### 4.6.2. Culture and Context

DEWE was thought to benefit from greater contextualization. Cultural adaptations were thought necessary to convey the unique challenges of dementia care in India. Greater emphasis on familial systems of support was considered the most relevant adaptation to the Indian context.


*“The context was good, but I think some of the tools which are available for people in probably UK settings are not available for people here, for example, like, xxxx shared about having a community of people with dementia and community helping each other so, we don’t have that kind of a network here”.*
[P8-02-01]

Contextualization of DEWE to the Indian context would also require content adapted to address unique cultural perspectives and challenges including regional language variations.


*“For Northern India must have to think about the Hindi language and their perspectives, and South India might have to think a little different…like that, we might have to add a few videos or something… so that it feels that, yes, it is our own problems here and it is what our people are going through”.*
[P5-01-01]

## 5. Results: Mixed Methods

Integration of quantitative and quantitative findings highlighted several motivating factors for engagement in DEWE including support with continuing professional development, access to expert facilitation, certification for learning, and education at no cost. The integrated findings also resulted in the identification of strengths and areas for improvements in DEWE (Table 4).

## 6. Discussion

Dementia training that is dependent on learner-initiated asynchronous instruction involving episodic or standardized elements of care often fails to accurately depict the trajectory of the illness and risks assumptions that such criteria are ubiquitous to the dementia syndrome [29]. One might also presume that such training has the potential to inadvertently introduce unconscious negative bias and reinforce stigma when certain training packages, e.g., Management of Stress & Distress, are recommended as mandatory dementia training. Such an approach unfortunately does not seem to help practitioners understand dementia as a neurodegenerative disorder and the complexities associated with the progression of the disease. Acknowledging the inherent uniqueness of the dementia trajectory for each person ensured that DEWE prioritized and enabled learning for participants to see the person behind the illness rather than the condition itself or focus on managing isolated behaviors associated with the condition [30]. DEWE, as both a unique and comprehensive dementia training program, used a “journey-based approach” to help participants to appreciate the essentials, priorities, and enablers for high-quality person and relationship-centered dementia care.

The COVID-19 pandemic prompted a shift in educational delivery and a transition to online programs which was particularly notable in India where e-learning has only started to gain popularity [31]. DEWE was developed during the pandemic to provide dementia education through a balance of flexibility (asynchronous workbooks) and immediacy (synchronous sessions). Previous research has demonstrated that the combination of asynchronous and synchronous (bichronous) online learning has several advantages including improved learning outcomes [17].

Asynchronous workbooks provided participants with access to dementia education at any time and place. Convenience and flexibility are benefits of technology-enabled dementia education [32,33,34]; however, asynchronous learning may result in feelings of isolation [35]. DEWE implemented various active learning strategies to compensate the effects of learner isolation. Adherence to principles of adult learning [36] combined with established motivating factors for participation, e.g., certification for learning [37], optimized learner interest and enthusiasm. Feedback is an integral element of teaching and learning across educational delivery modes [38,39]. Formal assessment of learning in DEWE may indicate whether specified learning outcomes were achieved and enhance learner motivation by fostering engagement with course content [40].

It is recommended that effective dementia education be relevant to the roles, experiences, and practices of learners, rather than adopting a one-size-fits-all approach [41]. However, diverse and multidisciplinary learning cohorts are also known to be beneficial [42]. DEWE replicated in-person teaching in synchronous sessions involving various practitioners and educators in real-time group discussions. This rich diversity of participants shared a passion for dementia and actively sought improvements in dementia care which led to a vibrant community of engagement and expertise [43].

Online programs should be designed to focus on pedagogical issues which emphasize collaborative and case-based learning (CBL) [31]. CBL aimed to connect theory and practice in both asynchronous and synchronous resources through emotional engagement with authentic clinical cases [44]. CBL using actual learner experience is aligned with the experiential learning model [45]. Greater emphasis on participants’ direct experience may develop higher order thinking whilst encouraging group reflection and collaborative problem solving [46]. Experience-based CBL during synchronous sessions could be enhanced using frameworks rooted in nursing processes to guide discussions and the effective articulation of cases [47].

DEWE was deemed highly relevant to the participants’ current practice; however, it fell short in addressing cultural differences between the UK and India. Both countries share many common goals, such as providing quality healthcare and addressing the needs of the growing elderly population [48]. Culturally appropriate education is crucial in India to address the rising burden of dementia and effectively address the unique socio-economic, cultural, linguistic, geographical, and lifestyle characteristics of this population [49]. Learner motivation can be enhanced where educational content includes references to national dementia prevalence, incidence, and related factors, along with information on local professional and community services [50]. Incorporating relevant material is crucial as participants anticipate applying new knowledge locally in practice.

DEWE was enriched through the involvement of individuals with lived experience of dementia and their family carers. Involvement in dementia education may foster a sense of worth and altruism among people with dementia and can provide carers with opportunities to share experiences and address caregiving challenges. The involvement of lived experience in DEWE fostered emotional connections with participants for deeper engagement [51]. Engagement was further enhanced from expert facilitation and input from credible and relatable dementia care experts who served as positive dementia care role models [52].

Surr [41] suggested that the total duration of dementia education programs should be more than eight hours with individual sessions being at least 90 min. Limited evidence exists on the optimal duration for online synchronous sessions; however, during or after these sessions, learners often report feelings of fatigue [53]. The average duration of synchronous sessions in DEWE was 82 min with most exceeding the 90 min recommendation and were regularly interspaced with breakout opportunities. Nonetheless, the duration of synchronous sessions in future iterations of DEWE might be minimized to mitigate cognitive overload and learner fatigue.

Delivery via WebEx posed challenges to participants from India due to an unfamiliarity with the platform. Engagement may have been further compromised as several participants used mobile phones to access DEWE. Mobile devices including smartphones are promising as learning devices due to their portability and ability to facilitate rapid information retrieval, collaborative interaction, and situated learning [54]; however, mobile devices may not be ideal for online learning due to software compatibility issues and small screen sizes [55].

Technical challenges and poor online etiquette can risk misunderstanding and miscommunication [56]. Setting rules for online etiquette in synchronous sessions may encourage a sense of community and active participation amongst learners [57]. In future courses, it will be important to provide ‘netiquette’ guidance in the first synchronous session, with contingency plans to resolve potential technical issues. These enhancements will ensure that DEWE provides a seamless and efficient online learning experience.

### Limitations

The survey did not include demographic information to establish the impact of the training based on country of origin. Social care practitioners who completed the survey consisted of care home and home care professionals; however, specific counts for the subgroups were not reported.

Most (78%) interview participants were from India which limits the transferability of the findings to the UK. This disparity also highlights the need for culturally sensitive learning content should DEWE be provided for learners based exclusively in the UK or India. Interviews were conducted up to 18 months post-training which allowed participants time to observe and comment on practice-based behavioral changes; however, this timeframe may also have introduced recall bias. The research team frequently discussed qualitative analytical procedures; however, the absence of respondent validation for interview transcripts may have compromised the methodological rigor of the inductive analytical process. The final sample size (n = 9) represented over a quarter of all DEWE participants that completed the online survey. Whilst practical, this approach did not adhere to robust principles of data saturation and potentially limited the emergence of additional insights and assurances of analytical rigor.

## 7. Conclusions

DEWE is an innovative resource and response to dementia education and training during the COVID-19 pandemic, integrating asynchronous and synchronous instruction. The person- and relationship-centered approach across the trajectory of the illness has relevance to diverse participants and educational settings which fosters an inclusive and supportive learning environment. Communities of engagement and expertise play a crucial role in enriching the overall learning experience. By bringing together participants with individuals with lived experiences and specialist knowledge, these communities can flourish to enhance learning and foster personal and professional growth. DEWE offers a notable advantage in its ability to overcome geographical boundaries; however, international dissemination will require cultural adaptations to ensure that the program aligns with cultural contexts for maximum impact. Future research will be useful to assess the effectiveness of DEWE using pre- and post-test methodologies or in comparative studies involving alternative modes of dementia education, including face-to-face or asynchronous instruction alone.

## Figures and Tables

**Figure 1 healthcare-12-00590-f001:**
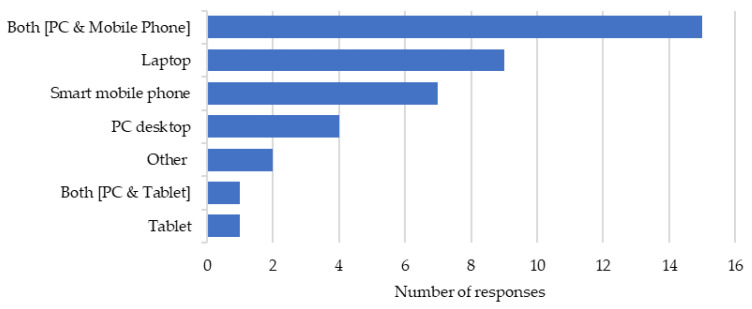
Devices used to access DEWE.

**Figure 2 healthcare-12-00590-f002:**
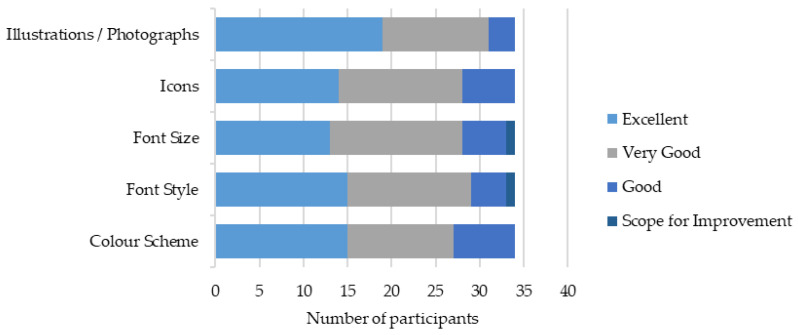
Presentation of the workbooks.

**Figure 3 healthcare-12-00590-f003:**
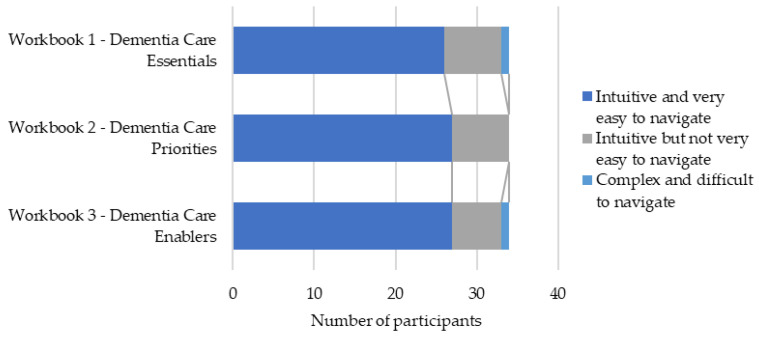
Ease of workbook navigation.

**Figure 4 healthcare-12-00590-f004:**
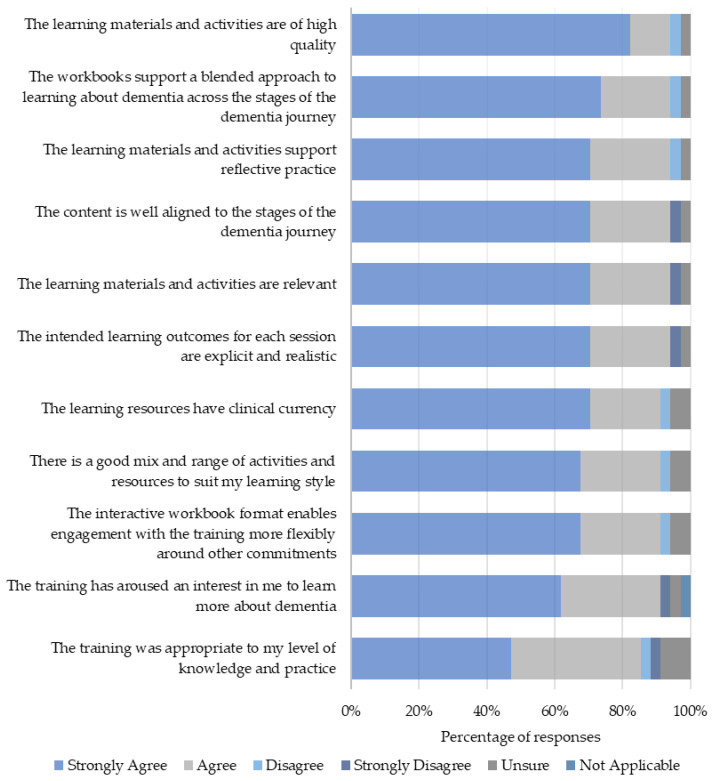
Reaction to the training content.

**Figure 5 healthcare-12-00590-f005:**
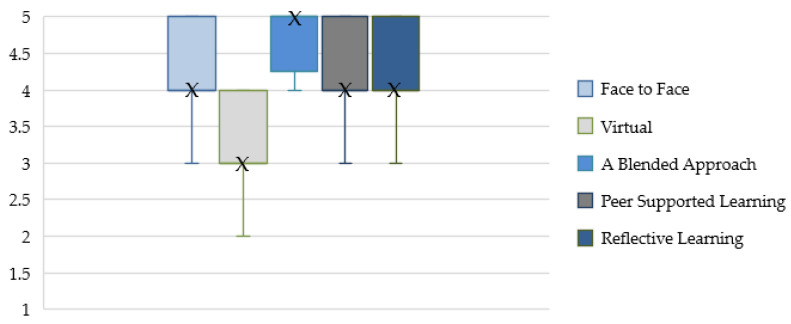
Preferred modes of dementia training. X denotes the median score.

**Figure 6 healthcare-12-00590-f006:**
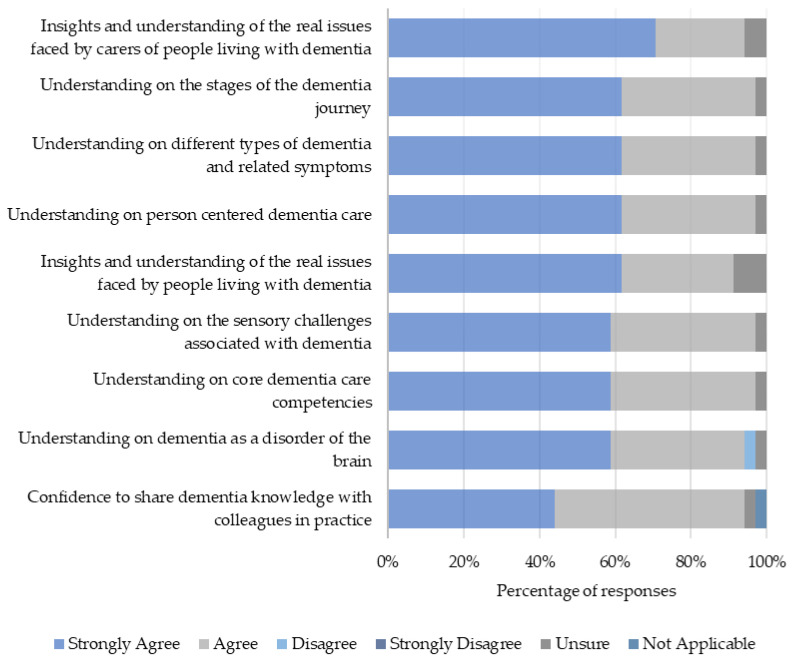
Learning gains following DEWE.

**Figure 7 healthcare-12-00590-f007:**
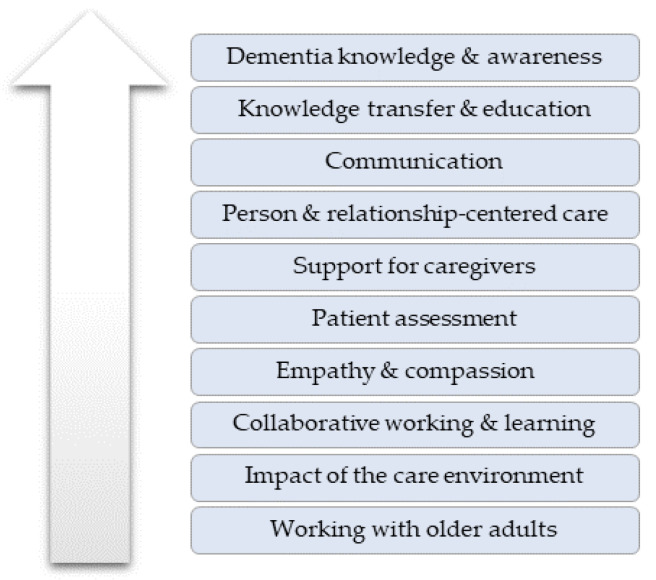
Transferable skills gained following DEWE.

**Figure 8 healthcare-12-00590-f008:**
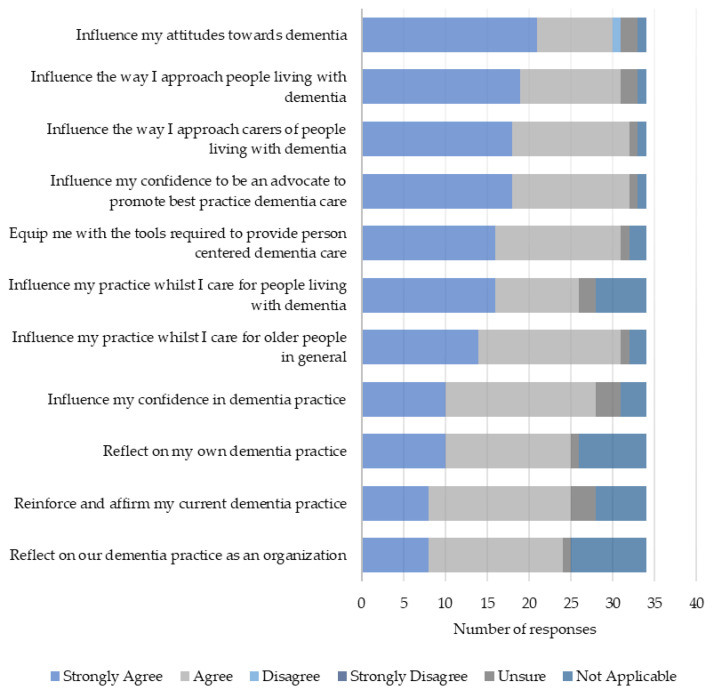
Practice-based behaviors following DEWE.

**Figure 9 healthcare-12-00590-f009:**
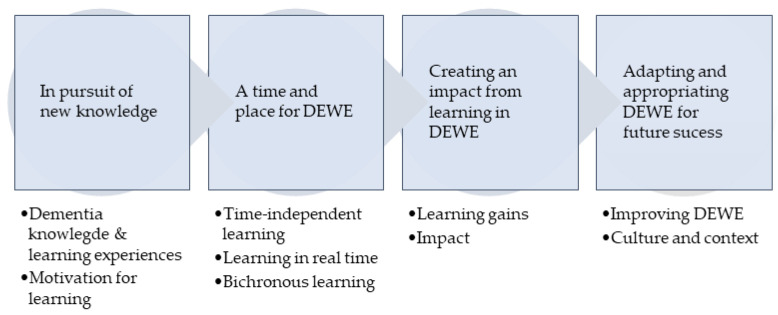
Themes and subthemes.

**Table 1 healthcare-12-00590-t001:** Themed content in DEWE.

Asynchronous		Synchronous		
Workbook		Session	Duration	Facilitator
	Introduction to DEWE and Workbooks	1	30 min	1
1. Dementia Care Essentials	The Human Brain	2	1 h	1
	Dementia: What it is and what it is not	3	1 h	1
	Memory Changes in Dementia	4	30 min	1
	Sensory Changes in Dementia	5	30 min	1
2. Dementia Care Priorities	Living with Dementia: Person-centered approaches	6	1 h	1
	Stages of the Dementia Journey (Barbara’s Story ^a^)	7	3 h	2
	Carer’s Perspectives in Dementia	8	1 h	3
	Stress & Distress in Dementia	9	1 h	1, 4
	Dementia, Delirium and Depression	10	2 h	2
3. Dementia Care Enablers	Staff: Self-care and wellbeing during COVID-19 ^b^	11	-	-
	Dementia & COVID-19 in Care Homes	12	1.5 h	5
	Implementing Change and Improvement	13	1.5 h	5
	Dementia Inclusive and Enabling Environments	14	2 h	1, 6
	End-of-life care in Dementia	15	2 h	1
	Relationship-centered Approaches to Dementia Care using the Six Senses Framework	16	2 h	1

^a^ Included five principles of the Adults with Incapacity Act, Scotland, 2000 [23]; ^b^ Self-directed learning using NES staff wellbeing resources [24]; Facilitator: (1) Senior Lecturer—Nursing (Dementia Expertise); (2) Dementia Nurse Consultant; (3) Person with lived experience of dementia (Family Carer); (4) Clinical Psychologist; (5) Senior Lecturer—Nursing (Quality Improvement Expertise); (6) Dementia Link Worker.

**Table 2 healthcare-12-00590-t002:** The aspects of DEWE most and least enjoyed by participants.

Enjoyed Most	Enjoyed Least
1. Group discussion and interactivity	1. Technical issues/videoconferencing platform
2. Course presentation/expert facilitation	2. Difficulty coordinating attendance
3. Reflective activities	3. Duration of interactive sessions (too long)
4. Workbooks	4. Duration of interactive sessions (too short)
5. Use of real-life stories and examples	5. Lack of participant interaction in break out rooms
6. Videos	6. Reflective activities
7. Structure and content (general)	7. Overall course duration (too short)
8. New knowledge	8. Digital skill requirements
9. Technology-enabled/blended approach	9. Technology-enabled delivery mode
10. Convenience	

**Table 3 healthcare-12-00590-t003:** Aspects of knowledge gained.

Knowledge Aspect	Quote
Types of dementia	*“We learned that there is like 200 different types of dementia…I had no idea, and I’ve got twenty plus years’ experience, like, in health and social care”.* [P9-02-02]
Neurophysiology	*“The role of hippocampus…so what is the difference between normal ageing and dementia, especially when we talk about memory changes…that is what is more important to know”.* [P6-01-01]
Stress & Distress	*“I really learned from this course…like analyzing a situation using the ABC chart…when something happens”.* [P5-01-01]
Delirium	*“Particularly in my clinical role, you know, in acute trauma, being able to identify things like delirium is so helpful, so those tools were really good”.* [P7-01-02]
Person-centered care	*“I have learned tools on how to handle different situations and how to understand why the person is behaving in a certain way and therefore how you kind of adapt your care to suit that person’s needs”.* [P8-02-01]
Relationship-centered care	*“I really liked the six senses approach, I thought that was really, really useful”.* [P7-01-02]
Care for caregivers	*“I understood that not only people living with dementia that needs attention, but we have to pay special attention even to the carers of people living with dementia”.* [P4-01-01]
Dignity & Respect	*“Taking your time with people, never getting cross, and the fact that you might have to answer the same thing…you’re answering the same questions but doing so every time with patience rather than frustration can make a huge difference”.* [P7-01-02]
Care environment	*“I remember some of the classes where they said how does the environment impact a person living with dementia, so, I think even the environment also is very, very important, the type of environment that we create around these people”.* [P4-01-01]
Care empathy	*“It can be very frustrating to have dementia and to place ourselves in their shoes, like some of those case studies we looked at, and listening to the various talks, that really helped me”.* [P1-01-01]

**Table 4 healthcare-12-00590-t004:** Summary of DEWE strengths and areas for improvement. The arrow up symbol denotes “increasing levels”.

	Strengths	Areas for Development
**Satisfaction**		
Asynchronous	High quality resources	Consider cultural adaptations
	Easy to use	Provide more assessments
	Time for learning	Improve placement of videos
	Learning on demand	Improve quality of resources in print form
	Active learning	Update content regularly
	Real-life videos	Consider strategies for learner motivation
	Preparation for synchronous sessions	
Synchronous	Interactive	Include interactive icebreakers session
	Active learning and reflection	*Discuss/mitigate technical issues*
	Diversity of participants	*Discuss/mitigate poor online etiquette*
	Includes dementia experts	Optimize session duration
	Includes people with lived experience	Optimize delivery platform
	Provides expert facilitation	Include more discussions on actual cases
	Consolidates asynchronous learning	Provide follow-up sessions
Bichronous	Highly rated	-
	Mutual learning between resources	
	Ongoing access to resources	
	Relevance post-COVID-19	
**Learning**	New knowledge	-
	New skills	
	Positive attitudes	
**Behaviors**	Positive behavior change	-
**Results**	↑ Dementia awareness	-
	↑ Knowledge transfer	
	↑ Development of educational resources	
	↑ Engagement with dementia education	
	↑ Engagement with dementia research	
	Dissemination of DEWE (wider audience)	

## Data Availability

The data that support the findings of this study are not openly available since participant consent was not gained for this purpose.

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
