# Peer review of "Dementia Education for Workforce Excellence: Evaluation of a Novel Bichronous Approach"

_healthcare, 2024, doi:10.3390/healthcare12050590_

Round 1
Reviewer 1 Report
Comments and Suggestions for Authors
This is a very interesting article with huge potential for awareness and education of nurses to clients with dementia. The aims were to explore the use of a learning approach and content (so pedagogical). The participants were to represent three different environments but it is not clear what these environments were. It also remains unclear why these participants are from Scotland and India - are these individuals in Scotland and India or all located within Scotland? The rationale for inclusion of India is unclear and also it is rather remiss that the cultural frame or issues were not considered and came out in the results and conclusions. The key here is clarity and rational not that it is inappropriate. The data is rather extensive and several tabll presented when a more summarised and analysed narrative would be useful.
Author Response
Thank you for your very helpful review. Please find below our responses to your review points:
This is a very interesting article with huge potential for awareness and education of nurses to clients with dementia. The aims were to explore the use of a learning approach and content (so pedagogical).
The participants were to represent three different environments, but it is not clear what these environments were.
A new section has been added (section 1.2) to make clear that participants represented two different environments (social care and nursing education). Social care participants included care home and home care practitioners; however, these subgroups were not delineated in the analysis. This has now been reported as a limitation.
It also remains unclear why these participants are from Scotland and India - are these individuals in Scotland and India or all located within Scotland?
The limitations section includes a paragraph to explain that the survey did not include demographic information to establish the findings based on country of origin. The country of origin of interview participants is reported (section 4.1).
The rationale for inclusion of India is unclear and also it is rather remiss that the cultural frame or issues were not considered and came out in the results and conclusions. The key here is clarity and rationale not that it is inappropriate.
The introduction section describes the need for dementia education in India which was intended to provide a brief rationale for including participants from India. The rationale is now further supported in new section 1.2 which further clarifies our rationale for delivering the training to participants from India.
The data is rather extensive, and several tables presented when a more summarised and analysed narrative would be useful.
Tables 2, 4, and 5 have been removed with the data provided in narrative summaries.
Reviewer 2 Report
Comments and Suggestions for Authors
The paper deals with a relevant topic in the global context. It proposes an education method with good strategies that can be incorporated in other sanitary crisis moments.
It is a well-written manuscript, which makes all the steps carried out clear, allowing the reproducibility of the study.
I suggest a review of the study limitations, since the study is well located, with a small sample, qualitative method, and it cannot be generalized as a condition of the country where it was developed. Adaptations may be necessary for the country itself.
Author Response
Thank you for your very helpful review. Please find below our responses to your review points:
The paper deals with a relevant topic in the global context. It proposes an education method with good strategies that can be incorporated in other sanitary crisis moments. It is a well-written manuscript, which makes all the steps carried out clear, allowing the reproducibility of the study.
I suggest a review of the study limitations, since the study is well located, with a small sample, qualitative method, and it cannot be generalized as a condition of the country where it was developed. Adaptations may be necessary for the country itself.
The study limitations have been reviewed and the issue of the small qualitative sample addressed. The limitation section now details how the findings cannot be generalized to the UK context and highlights the need for potential cultural adaptations.
Reviewer 3 Report
Comments and Suggestions for Authors
This study make use of a robust conceptual framework and adhering to standards in data collection and analysis. However, elaboration on sampling strategies and additional information pertaining to the survey and interview instruments will be helpful.
Furthermore, an explanation on data nuances, integration and thematic procedures would be helpful to the reader.
Author Response
Thank you for your very helpful review. Please find below our responses to your review points:
This study make use of a robust conceptual framework and adhering to standards in data collection and analysis.
However, elaboration on sampling strategies and additional information pertaining to the survey and interview instruments will be helpful.
A new section (2.1) has been added to elaborate on sampling and recruitment. Specifically, the section details that all DEWE participants were invited to take part in the online survey and invited to take part in interviews. Section 2.2 now provides information about the online survey platform and the nature of the data collected including how the survey was constructed, i.e., according to Kirkpatrick’s Model. The authors conclude that there is not much more to add about the survey since the content is made quite explicit in the reporting of findings including comprehensive tables and figures (e.g., figs 1-8). The interview guide will be included as supplementary material as indicated in section 2.2.
Furthermore, an explanation on data nuances, integration and thematic procedures would be helpful to the reader.
Section 2.3 has been updated to provide a more nuanced account of the data analysis and to provide a more detailed account of the methods involved in survey and interview data integration. The authors believe that section 2.3 provides a sufficient account of the 6 stages involved in the thematic analytic procedures. Limitations to the thematic procedures have been identified and reported in section 6.1.
Reviewer 4 Report
Comments and Suggestions for Authors
Dear authors. I find the subject of your article, as well as the methodology used, very interesting, with a very pertinent public health theme and focused on increasingly frequent and current learning methods.
I think the article could eventually be published after a few revisions. My suggestions for improvement:
Materials and Methods (page 3): Regarding the data collection component, the authors should clarify why they recruited participants to complete a questionnaire at 6 months and interviews in a time window of up to 18 months after participation.
“ (…) Quantitative and free-text response data were collected via an online survey platform 116 [26] six months after participation in DEWE using a variety of Likert scale, rating scale, 117 and open-ended questions. Consent was gained from all participants using the online sur- 118 vey platform. 119 Nine semi-structured interviews were conducted virtually via the Cisco Webex plat- 120 form between January and February 2023 (within 18 months of participation in DEWE).”
Materials and Methods (page 4): considering the data analysis presented, the authors do not show evidence of the verifiability of the data, namely the return to the participants of the interview after its transcription or even their active participation with feedback on the results found by the research team. This aspect diminishes the methodological quality of the inductive process and should be addressed at this stage and added as a limitation of the research.
Table 2 (page 4): The information contained in table 2 is sparse, it could be summarized in one sentence as there are only two variables evaluated, thus dispensing the need of this table.
Figures 2, 3 and 4 (pages 6 and 7): in order to make it easier for readers to interpret the data, the graphs should be drawn up along the same lines. The authors have a vertical orientation in graphs 2 and 4 and a horizontal orientation in graph 3; this should be standardized.
Table 4 (page 9): With regard to table 4, it needs to be clarified what the authors' aim is in presenting it. If the participants answered that the aspects were all relevant and the table only shows 2 descriptive statistics, this could be transposed into a sentence and eliminate this table, which does not add relevant information.
Results (page 12): Regarding Demographics presentation, table 5 can also be eliminated; the authors say it resumes the data but in fact it occupies more space than the paragraph before and only adds two information’s: country and profession of interviewed participants.
Limitations (page 20): The authors must refer to the methodological issue of data verifiability as a limitation, as they did not confront the participants with the transcribed interview or the discussion of results. In addition, they should address the issue of saturation of qualitative data, as it was achieved with 9 interviews.
Author Response
Thank you for your very helpful review. Please find below our responses to your review points:
Materials and Methods (page 3): Regarding the data collection component, the authors should clarify why they recruited participants to complete a questionnaire at 6 months and interviews in a time window of up to 18 months after participation.
Clarification has been provided (section 2.2).
Materials and Methods (page 4): considering the data analysis presented, the authors do not show evidence of the verifiability of the data, namely the return to the participants of the interview after its transcription or even their active participation with feedback on the results found by the research team. This aspect diminishes the methodological quality of the inductive process and should be addressed at this stage and added as a limitation of the research.
The authors acknowledge the significance of demonstrating data verifiability to enhance the methodological rigor of the research. However, since this was not considered during the conceptualisation phase, the authors would suggest that this is more suitable to address it as a limitation (see below).
Table 2 (page 4): The information contained in table 2 is sparse, it could be summarized in one sentence as there are only two variables evaluated, thus dispensing the need of this table.
Table 2 removed with content provided in a narrative summary.
Figures 2, 3 and 4 (pages 6 and 7): in order to make it easier for readers to interpret the data, the graphs should be drawn up along the same lines. The authors have a vertical orientation in graphs 2 and 4 and a horizontal orientation in graph 3; this should be standardized.
The orientation of figure 2 has been changed for consistency with figures 1, 3, and 4.
Table 4 (page 9): With regard to table 4, it needs to be clarified what the authors' aim is in presenting it. If the participants answered that the aspects were all relevant and the table only shows 2 descriptive statistics, this could be transposed into a sentence and eliminate this table, which does not add relevant information.
Table 4 eliminated with content transposed into a sentence.
Results (page 12): Regarding Demographics presentation, table 5 can also be eliminated; the authors say it resumes the data but in fact it occupies more space than the paragraph before and only adds two information’s: country and profession of interviewed participants.
Table 5 eliminated.
Limitations (page 20): The authors must refer to the methodological issue of data verifiability as a limitation, as they did not confront the participants with the transcribed interview or the discussion of results. In addition, they should address the issue of saturation of qualitative data, as it was achieved with 9 interviews.
Data verifiability and issues of qualitative sample size and data saturation have been addressed as limitations.